# Anti-Cancer and Anti-Inflammatory Activities of a Short Molecule, PS14 Derived from the Virulent Cellulose Binding Domain of *Aphanomyces invadans*, on Human Laryngeal Epithelial Cells and an In Vivo Zebrafish Embryo Model

**DOI:** 10.3390/molecules27217333

**Published:** 2022-10-28

**Authors:** Manikandan Velayutham, Purabi Sarkar, Gokul Sudhakaran, Khalid Abdullah Al-Ghanim, Shahid Maboob, Annie Juliet, Ajay Guru, Saravanan Muthupandian, Jesu Arockiaraj

**Affiliations:** 1Department of Biotechnology, College of Science and Humanities, SRM Institute of Science and Technology, Chennai 603 203, Tamil Nadu, India; 2Department of Molecular Medicine, School of Allied Healthcare and Sciences, Jain Deemed-to-be University, Bangalore 560 066, Karnataka, India; 3Department of Zoology, College of Science, King Saud University, Riyadh 11451, Saudi Arabia; 4Institute for Cellular and Molecular Biology, The University of Texas at Austin, Austin, TX 78712, USA; 5Department of Conservative Dentistry and Endodontics, Saveetha Dental College and Hospitals, SIMATS, Chennai 600 077, Tamil Nadu, India; 6AMR and Nanomedicine Lab, Department of Pharmacology, Saveetha Dental College, Saveetha Institute of Medical and Technical Sciencess (SIMATS), Chennai 600 077, Tamil Nadu, India

**Keywords:** cellulase binding domain, zebrafish, virulent molecule, anti-cancer, ant-inflammatory

## Abstract

In this study, the anti-cancer and anti-inflammatory activities of PS14, a short peptide derived from the cellulase binding domain of pathogenic fungus, *Aphanomyces invadans*, have been evaluated, in vitro and in vivo. Bioinformatics analysis of PS14 revealed the physicochemical properties and the web-based predictions, which indicate that PS14 is non-toxic, and it has the potential to elicit anti-cancer and anti-inflammatory activities. These in silico results were experimentally validated through in vitro (L6 or Hep-2 cells) and in vivo (zebrafish embryo or larvae) models. Experimental results showed that PS14 is non-toxic in L6 cells and the zebrafish embryo, and it elicits an antitumor effect Hep-2 cells and zebrafish embryos. Anticancer activity assays, in terms of MTT, trypan blue and LDH assays, showed a dose-dependent inhibitory effect on cell proliferation. Moreover, in the epithelial cancer cells and zebrafish embryos, the peptide challenge (i) caused significant changes in the cytomorphology and induced apoptosis; (ii) triggered ROS generation; and (iii) showed a significant up-regulation of anti-cancer genes including BAX, Caspase 3, Caspase 9 and down-regulation of Bcl-2, in vitro. The anti-inflammatory activity of PS14 was observed in the cell-free in vitro assays for the inhibition of proteinase and lipoxygenase, and heat-induced hemolysis and hypotonicity-induced hemolysis. Together, this study has identified that PS14 has anti-cancer and anti-inflammatory activities, while being non-toxic, in vitro and in vivo. Future experiments can focus on the clinical or pharmacodynamics aspects of PS14.

## 1. Introduction

Cancer is the second leading cause of death before 70 years of age in 183 countries, as per the World Health Organization (WHO) [1,2]. Laryngeal carcinoma is the second most common form of head and neck cancer. Early detection and appropriate preoperative evaluation enhance the success rate of the anti-cancer therapy. Patients with initial-stage (T1 and T2) cancers have successful results, with a recovery rate between 80 and 90% [3,4]. Laryngotomy is an important therapeutic possibility in treating laryngeal cancer. However, this therapy is expensive and has critical side effects, including the onset of hypothyroidism [5]. Therefore, an efficient anti-cancer therapy is required, that which could induce apoptosis, the intended role of any successful cancer treatment. Inflammation is a physiological event in the body in response to injury or infection. An intricate network of several molecules regulates the human immunological and inflammatory responses. Transfer of cells and components, such as leukocytes, antibodies and cytokines, among others, via circulation, to the site of injury, infection or immunological reactivity, is induced by an increase in blood flow and capillary permeability [6].

Of late, bioactive peptides have been considered as efficient pharmacological molecules for anti-cancer therapeutics. Numerous reports demonstrate that the peptides derived from natural and animal sources show several biofunctional activity, such as immunomodulatory, antibacterial, antihypertensive, antithrombotic, anti-cancer, antioxidative and cholesterol-lowering properties [7]. Anti-cancer peptides (ACP) are molecularly targeted peptides that can penetrate and directly connect to the membranes of specific cancerous cells, organelles or binding peptides that deliver the anti-cancer medicines [8]. Depending on the number of amino acid residues, sequence length, composition, molecular weight, isoelectric point, net charge, hydrophobicity, amphiphilicity, secondary structure and structural orientation, peptides can be used as molecularly targeted drugs and ‘guiding missiles’ to inhibit cell proliferation or completely eliminate cancer cells [9]. Internal prolines in peptides, similar to glycine residues, are important for membrane contact and conformational flexibility. Serine and glycine residues have been shown to reduce tumor growth and improve anti-proliferative effects. Methionine, a mildly hydrophobic amino acid, does not have a significant function as an ACP, although it can be absorbed by cancer cells in large quantities. Furthermore, a methionine-deficient diet produces a metabolic abnormality in cancer cells by halting their growth. Tyrosine and tryptophan residues are mildly hydrophobic amino acids; tyrosine does not affect ACP toxicity; however, tryptophan may affect ACP toxicity against cancer cells. Synthesized peptides containing tryptophan and histidine, on the other hand, may reduce cytotoxicity [10].

The virulent molecule, cellulose binding domine (CBD) was identified from the earlier constructed transcriptome dataset of epizootic ulcerative syndrome (EUS) causing pathogenic fungus, *Aphanomyces invadans* [11]. Cellulose is the most prevalent biopolymer, and it elicits a crucial function in the structure of plants. Fungi, bacteria and algae have produced a group of enzymes (belonging to the CBD-carrying cellulosome-integrating protein) that preferentially degrade cellulose and convert it to sugars that the organism may conveniently consume as food [12]. Cellulolytic enzymes can be linked to form multi-enzymatic complexes (cellulosomes) or it can exist independently as distinct enzymes; enzymes have a modular structure in both circumstances. Unassociated enzymes have a catalytic domain that catalyzes the hydrolysis reaction and a cellulose-binding domain that mediates the enzymes’ binding to the substrate [13,14]. A linker peptide connects the two domains and must be long and flexible enough to facilitate efficient orientation and operation of both domains. The cellulosome enzymes are noncovalently attached to the CBD-carrying cellulosome-integrating protein [15]. In this study, we have characterized the physicochemical properties of PS14 of CBD protein. Toxicity of PS14 was accessed using in vitro L6 cells and an in vivo zebrafish larval model. Moreover, the peptide’s potential for anti-cancer and anti-inflammatory activity has been evaluated.

## 2. Results and Discussion

### 2.1. CBD Protein and PS14 Analysis

The identified full-length cDNA sequence encodes 267 amino acids. BLAST analysis showed that the CBD protein sequence has 100% identity with a hypothetical protein (H310_09802) of *A. Invadans*. This hypothetical protein has three fungal cellulose-binding domains; hence, this protein of interest is indeed a cellulase binding domain (CBD) protein. The chemical formula of the protein was C_1241_H_1891_N_333_O_397S21_, its molecular mass was 28500.92 gm/mol and its theoretical pI was 4.54. In the CBD protein, there were 23 negatively charged residues (Asp + Glu), and 12 positively charged residues (Arg + Lys); and the estimated half-life of CBD protein was 30 h (mammalian reticulocytes, in vitro). Moreover, the instability index of the CBD protein was 28.98, aliphatic index was 66.48 and the grand average of hydropathicity (GRAVY) was −0.151. The 3D structural view of the protein is provided as Appendix A. Previous studies indicate that proteins and peptides of pathogenic fungal origin have anti-cancer potential. For example, metabolites or small molecules from fungus, *Aspergillus fumigatus* has anti-cancer potential; these small molecules are responsible for pathogenicity as well as the virulence nature of the fungus, *A. fumugatus* [16]. Previous reports [17,18] indicate that pathogenic fungus-derived virulence factors and the venom of numerous species including bees, snakes, scorpions and plant-derived toxins have significant anti-cancer activities. In line with this context, we hypothesized that the CBD protein-derived PS14 peptide could have significant anti-cancer potential.

The short molecule or the peptide, PS14 of CBD protein contained 14 amino acids residues as: ^NH2^Pro–Thr–Glu–Cys–Gln–Val–Gly–Thr–Thr–Cys–Lys–Val–Glu–Ser^COOH^. The physico-chemical properties of PS14 were as follows: molecular mass, 1481.65 g/mol; iso-electric point, 4.15; net charge at pH 7, −1.1; hydrophobicity, 0.321; polar residues, 9; non-polar residues, 5; hydrophobic moment, 0.093; uncharged residues, 6; charged residues, 3; special residues, 3; along with these characters, PS14 showed good water solubility. The 3D structure of the peptide and its helical wheel projection is shown in Appendix A. The anti-cancer property of PS14 was determined by the number of amino acid residues, combination of amino acids, molecular weight, net charge, hydrophobicity, hydrophilicity and amphipathic nature [10]. These observations suggest that PS14 meets the physicochemical criteria to be eligible for an anticancer peptide. In silico toxicity analysis of PS14 in ToxIBTL program revealed that PS14 is non-toxic and its score of 1.684647 × 10^−9^ was within the limit [19].

The mACPpred algorithm showed the anti-cancer probability score of PS14 was 0.626. The probability score of any anti-cancer peptide could be in the range between 0.0 and 1.0; PS14 falls within this range [20]. Hence, further anti-cancer assays using PS14 were performed in Hep-2 cells, which are derived from epidermoid carcinoma of the larynx. In addition, in silico analyses showed that PS14 has anti-inflammatory activity. This outcome was predicted using PreAIP, wherein the score for PS14 is 0.443, which reveals a medium confidence score on the anti-inflammatory activity. However, its sensitivity and specificity range was predicted to be 71.9% and 80.1%, respectively.

### 2.2. Assessment of PS14 for Toxicity

Toxicity of PS14 was evaluated using the MTT assay, in which the percentage of cell (L6 cells) viability was calculated. The dose-dependent cell viability of PS14-treated cells are as follows: 5 µM, 94.4 ± 2.66%; 10 µM, 87.31 ± 4%; 15 µM, 87.25 ± 7.21%; 20 µM, 84.44 ± 3.89% and 25 µM, 89.57± 2.39% (Figure 1). The Triton X-100 positive control showed 21.48 ± 2.7% viable cells. One of the significant properties of this anti-cancer peptide, PS14, is its specificity, to target the cancer cells and has no toxic effect on normal cells [10]. These results indicate that PS14 is non-toxic to L6 cells.

Recently, the zebrafish model has gained the attention of researchers due to its advantages in the phenotype and genotype. The zebrafish genome shares 85% sequence similarity with the human genome; it is a cost-effective model. This species has a short life cycle and is an easy model to administer/evaluate the test drugs [21]. Thus, in this study, zebrafish was used as an in vivo model for toxicity analysis. The hatching rate of PS14 exposed group was 97.53 ± 1.9% at 5 µM followed by 93.47 ± 1.5% at 10 µM, 91.78 ± 1.7% at 15 µM, 89.43 ± 1.9% at 20 µM and 89.14 ± 1.5% at 25 µM. Meanwhile, the positive control (H_2_O_2_) showed 37.25 ± 2% hatching rate (Appendix A). The positive control group showed a significant (*p <* 0.0) reduction in the hatching rate compared to the control. The concentration of PS14 at 20 and 25 µM showed slight variation in the hatching rates, but was inconsequential. The mortality rate and survival rate of the experimental group were calculated in 96 h post-fertilized (hpf) embryos. When compared with the untreated control, the positive control (H_2_O_2_) showed a significant reduction in survival rate and increased death rate (Appendix A), but this was not in the case of PS14. In addition, the heart rate of the zebrafish larvae was not influenced by PS14 (Appendix A). Morphological characteristics of the 72 to 96 hpf larvae, exposed to PS14, were observed under the microscope. There were no abnormalities, in both the groups, control or PS14-treated group. However, in the positive control group, several embryos were recorded with abnormal morphological features, including bent spin and tail (Figure 2).

PS14 exposure to zebrafish showed no signs of developmental toxicity. This observation was further evaluated by the multiple fluorescent staining assays. The DCFDA staining approach measured the intercellular ROS concentration. The non-fluorescent DCFDA reacts with the intracellular free radicals to be converted to the fluorescent DCF. Fluorescence intensity was directly proportional to the intracellular ROS concentration [22] as shown in Figure 3A. The untreated control showed relatively less fluorescence intensity, while the H_2_O_2_-treated positive control showed the highest fluorescent intensity. The PS14 challenge, at different peptide concentrations, accordingly, showed variation in the fluorescence intensity. Results, in terms of different fluorescence intensities, were quantified by processing the pictographic data using the Image J software, as shown in Appendix A.

The acridine orange assay identified whether or not the PS14 challenge causes apoptosis. PS14 treatments, except at 25 µM PS14, did not induce any apoptotic cell death in zebrafish larvae. When compared with the H_2_O_2_ positive control, there was a significant change in the fluorescence intensity in the 25 µM PS14 challenge, as shown in Figure 3B. This outcome has been verified further for its correctness, using the data quantification on fluorescent intensity using image J Software (Ver.1.49; NIH, Bethesda, MD, USA) (Appendix A). Interestingly, this result correlated with our previous finding, wherein GR15, a peptide from S-adenosylmethionine synthase of cyanobacteria produced significant protective effect against H_2_O_2_-induced ROS [23]_._ Similar to GR15, PS14 exposure showed a reduction in fluorescent intensity, which shows that PS14 did not induce apoptosis in the zebrafish larvae.

### 2.3. In Vitro Anti-Cancer Activity of PS14 against Hep-2 Cells

The anti-cancer potential of PS14 and its efficiency to inhibit the Hep-2 cell proliferation within 24 and 48 h, dose dependently, has been determined by the MTT assay (Table 1). The concentration of PS14 which inhibits 50% of cell proliferation (IC50) at 24 h was 21 µM, and at 48 h, it was 18.75 µM. We further confirmed the inhibition by counting the viable and dead cells through Trypan blue staining at 24 h and calculated the percentage. As predicted through bioinformatics, PS14 specifically targets the cancer cells rather than the non-cancerous cells. Chiangjong et al. [10] reported that the amino acid residues including Glu, Cys, Gly and Lys have anti-cancer properties; PS14 does have such amino acids in its sequence. An earlier study also found similar results, wherein MP12 peptide, from fungus virulent molecule cysteine-rich trypsin inhibitor showed significant anti-cancer properties against Hep-2 cells [8].

To determine the anticancer activity of PS14, total LDH release was evaluated. When cellular damage occurs, in response to it, LDH gets released to the growth medium because of the increase in the pore size of the cell membrane [24]. PS14 treatment showed a dose-dependent increase in the concentration of LDH release than the control. Similarly, a peptide named Brevilaterin B, which was derived from *Brevibacillus laterosporus*, has been shown to cause cell membrane damage, which leads to increased LDH release to the medium [25]. Based on these reported observations and the outcomes from PS14 challenge, it is evident that PS14 damages the Hep-2 cell membrane.

Damages in the cell membrane clearly affects cellular morphology. Abnormal morphology indicates apoptosis, such as granulation cell shrinkage [26]. PS14-treated cells showed loss of cell structure, granulation and shrinkage (Figure 4). Similar to this outcome, serine-threonine protein kinase protein of *Channa striatus* derived peptide IW13 showed anti-cancer activity against A549, MCF-7 and Hela cells. Moreover, IW13 significantly altered the cellular morphology [26]. Different apoptotic cell stages were identified by nuclear staining (Hoechst 33342). Apoptotic cells also showed different nuclear morphology, such as chromatin fragmentation, nuclear swelling, bi-or multi-nucleation, and chromatin condensation [27]. PS14-treated cells do exhibit such apoptotic features under the bright blue intensity of nuclear staining in a dose-dependent manner (Figure 5). Similarly, Wu et al. [28] reported that the *Anthopleura anjunae* oligopeptides showed significant changes in prostate cancer (DU-145) cells.

ROS production was investigated though DCFDA staining assay in PS14-treated Hep-2 cells. Results showed that the peptide exposure caused dose-dependent changes in the intracellular ROS generation (Figure 6), which was further assessed in the Image J program (Appendix A). Compared with the control group, there was a significant increase in the ROS concentration in the PS14-exposed group. Karanam and Arumugam [29] have reported a similar observation, wherein a dipeptide from *Callyspongia fstularis* (marine sponge) symbiont *Bacillus pumilus* AMK1 showed a significant increase in the level of ROS generation.

Typically, the antitumor drugs cause an increase in ROS generation, which leads to DNA damage and apoptosis by lowering the mitochondrial membrane potential. These changes eventually led to the release of cytochrome C, activation of Caspase 3 and increased Bax expression, and decreased Bcl2 expression [30]. In line with these observation, mRNA expression analyses in this study showed that PS14 at 25 µM concentration significantly up-regulated the BAX (2.56 ± 0.25-fold), Caspase-3 (3.25 ± 0.42-fold), Caspase-9 (4.8 ± 0.36-fold) expression and down-regulated Bcl-2 (0.56 ± 0.21-fold) (Figure 7).

### 2.4. Anti-Inflammatory Activity of PS14

The association between cancer and inflammation is widely accepted; wherein inflammatory cells in the tumour environment often accelerate tumorigenesis; and that the inflammatory cells and facilitators are core supporters of the cancer cells [31]. In this study, as predicted by the PreAIP, PS14 exhibits anti-inflammatory activity. Hence, the anti-inflammatory activity of PS14 was evaluated by the following experiments: proteinase inhibition assay, Lipoxygenases, heat-induced hemolysis and hypotonicity-induced hemolysis (Figure 8). Results of the proteinase inhibition assay showed that PS14, dose-dependently and significantly increased the proteinase inhibition, as: 25.45 ± 1% (5 µM), 39.21 ± 1.5% (10 µM), 49.10 ± 1.89% (15 µM), 69 ± 1.3% (20 µM) and 77 ± 2.8% (25 µM). These results are significantly (*p <* 0.05) different than the control. Lipoxygenases are the key enzyme that play a vital role in inflammatory response in human. This enzyme plays a critical role in triggering the inflammatory responses, under excess-ROS generation. PS14 challenge showed a dose-dependent inhibition at 5 µM (19.71 ± 1%) followed by the other higher concentration and the results are significantly different from the control (*p <* 0.05). In addition to these effects, PS14 dose-dependently inhibits both the heat-induced and hypotonicity-induced hemolysis.

## 3. Materials and Methods

### 3.1. Bioinformatic Investigation and Synthesis of PS14

The cDNA sequence of CBD has been identified from the transcriptomic data of *A. invadans* [11]. Identified cDNA sequence has been converted into the protein sequence using the ExPASy tool (http://web.expasy.org/translate/) (accessed on 2 January 2022) [32]. The protein sequence was then analyzed in NCBI BLAST for homology study; in addition, the physiochemical characterization of the protein was performed using Expasy’s ProtParam web server (http://us.expasy.org/tools/protparam.html) (accessed on 13 January 2022) [33]. The structural elucidation of CBD was conducted using I-TASSER online server; further, the structure was visualized in PyMol tool (Version 0.99). The HeliQuest online tool predicted the peptide from the protein and its physicochemical properties (http://heliquest.ipmc.cnrs.fr/cgi-bin/ComputParams.py) (accessed on 28 January 2022) [34]. The biological properties of PS14 were analyzed using the following programs: Toxicity was predicted atToxIBTL online tool (http://server.wei-group.net/ToxIBTL) (accessed on 5 February 2022) [35], anti–cancer property was screened at mACPpred algorithm (http://www.thegleelab.org/mACPpred/ACPEx ample.html) (accessed on 11 February 2022) [36], anti-inflammatory property was predicated at PreAIP (Prediction of Anti-Inflammatory Peptides) (http://kurata14.bio.kyutech.ac.jp/PreAIP/) (accessed on 22 February 2022) [37]. PS14 was synthesised at Zhengzhou Peptides Pharmaceutical Technology Co. Ltd., China, and its purity was checked in HPLC coupled with mass spectrometry and certified by the supplier, which is >95%. The synthesized PS14 was diluted in phosphate buffer saline (PBS) stock at 1mM concentration. The experimental concentration was prepared in PBS and the stock was stored at −20 °C.

### 3.2. Assessment of PS14-Induced Toxicity In Vitro and In Vivo

#### 3.2.1. Cell Culture and Maintenance

Rat skeletal myoblast cells (L6 cell line) and human laryngeal cancer cells (Hep-2) were obtained from the Department of Virology, The King Institute of Preventive Medicine and Research, Guindy, Chennai, India. The cells were cultured in a medium containing 10% fetal bovine serum (Gibco, Sydney, Australia), 1% of antibiotic-antimycotic solution (Sigma, St. Louis, MO, USA) and DMEM high glucose (4.5 g/L) (Sigma). The cells were maintained in a 5% CO_2_ incubator at 37 °C.

#### 3.2.2. Cytotoxicity Assessment of PS14 on L6 Cells

Cytotoxicity of PS14 was determined using the (4,5-dimethylthiazol-2-yl)-2,5-diphenyltetrazolium bromide assay (MTT) [38]. The L6 cells, cultured in a 96-well plate for 24 h at 37 °C in a 5% CO_2_ incubator, were treated with PS14 at various concentrations (5, 10, 15, 20 and 25 µM); the untreated cells were used as control. Triton X-100-treated cells were used as a positive control. Post-treatment, cells were incubated for 24 h at 37 °C and 5% CO_2_. After 24 h, 20 µL of 0.5 mg/mL MTT solution was added to each well, followed by 4 h of incubation, and the formazan crystals were dissolved by adding 200 µL of DMSO. An enzyme-linked immunosorbent assay (ELISA) reader was used to measure the absorbance at 570 nm to calculate the cell viability as follows and is presented in percentage:$$\%\;cell\;viability=\left[\frac{OD\;of\;treated\;cells}{OD\;of\;untreated\;control}\right]\times 100$$

#### 3.2.3. In Vivo Toxicity Assessment in Zebrafish Embryo

The wild-type adult zebrafish, *Danio rerio* was purchased from the NSK aquarium, Kolathur, Tamil Nadu, India. Maintenance of zebrafish, breeding and collection of embryos was carried out by following the standard procedures, as mentioned in our earlier report [38]. Toxicity assays using PS14 were executed with the 4 h post-fertilized (hpf) zebrafish embryos, which were grouped into seven categories; each group contained 30 embryos and maintained in a six-well plate containing the embryo media (3 mL). The embryos were treated with PS14 at different doses (5, 10, 15, 20 and 25 µM). The untreated group served as the control; while the embryos exposed to H_2_O_2_ (1 mM) were a positive control. Between 4 and 96 hpf, embryos were carefully examined for toxicological parameters including hatching rate, survival rate, mortality, heart rate and morphological abnormalities.

#### 3.2.4. Assessment of In Vivo ROS Production

In the zebrafish larvae that were exposed to PS14, ROS generation was evaluated using the 2,7-dichloro dihydro fluorescein diacetate (DCFH-DA) assay [38]. From the in vivo toxicity experimental group, three larvae (96 hpf) were randomly taken from each group. The larvae were euthanized, fixed in 4% paraformaldehyde and stained with 20 µg/mL DCFDA for 20 min in the dark. Stained embryos were washed with PBS and observed under a fluorescence microscope equipped with a Cool SNAP-Pro color digital camera (Olympus, Tokyo, Japan). Results were recorded as photomicrographs, and the fluorescence intensity was quantified using ImageJ software (V.1.49, NIH, Bethesda, MD, USA).

#### 3.2.5. Assessment of Apoptosis, In Vivo

Whether or not the PS14-exposed zebrafish larvae undergo apoptosis has been identified using the acridine orange (AO) staining assay [38]. As described elsewhere, selected larvae from the in vivo toxicity assay were stained using 7 µg/mL acridine orange and incubated in the dark at 28 °C for 20 min. They were then washed with PBS and observed under a fluorescence microscope facilitated with a Cool SNAP-Pro color digital camera (Olympus, Tokyo, Japan). Results were recorded as photomicrographs, and the fluorescence intensity was quantified using ImageJ software (V.1.49, NIH, USA).

### 3.3. In Vitro Anti-Cancer Activity of PS14

#### 3.3.1. Assessment of Anti-Proliferative Activity

The potential of PS14′s anti-proliferative activity was determined using the MTT assay [39] and Hep-2 cells. In brief, 1 × 10^6^ cells/well were seeded in a 96-well plate and incubated in a 5% CO_2_ incubator at 37 °C for 24 h. Cells were treated with the following five concentrations of PS14: 5, 10, 15, 20 and 25 µM, for 24 to 48 h. After treatment, 20 µL of MTT solution (5 mg/mL in sterile PBS) was added and incubated for 4 h. Formazan crystals were dissolved in 200 µL of DMSO (0.01%). Absorbance was measured at 570 nm in an ELISA reader (Multiskan Go ELISA reader, ThermoScientific, Vantaa, Finland).The effective anti-cancer activity was determined by calculating the IC_50_ value (the concentration that caused 50% cell death) from the data obtained from the inhibition rate using GraphPad Prism software.

#### 3.3.2. Trypan Blue Staining

Trypan blue staining assay validated the anti-cancer activity by determining the number of viable or dead cells, post PS14-challenge. In brief, Hep-2 cells were seeded in a six-well plate and allowed to reach the desired confluence. Cells were then treated with PS14 at various concentrations (5, 10, 15, 20 and 25 µM) for 24 h. The untreated group served as control. Treated cells were harvested by trypsinization, washed with PBS and stained with trypan blue. Stained cells were loaded in a hemocytometer, and observed in a light microscope to calculate the percentage of viable or dead cells [40].

#### 3.3.3. LDH Assay

Cytotoxicityof PS14 in Hep-2 cells was detected using the LDH assay. Cells were treated for 24 h with various concentrations of PS14 (5, 10, 15, 20 and 25 µM). The untreated cells were considered as control, whereas Triton X-100-treated cells were considered as a positive control. Post-treatment, the cells were aspirated and centrifuged at 1200× *g* rpm. To the reaction mixture, 0.20 mM NADH, 61.43 mMol Tris buffer pH 7.4 were added, incubated for 15 min and the absorbance was measured at 339 nm using an ELISA reader. The percentage of LDH release was calculated [41,42].

#### 3.3.4. Morphological Analysis and Apoptosis Staining

The effect of PS14 on cytomorphology was investigated using the peptide-treated Hep-2 cells [39]. In brief, Hep-2 cells were treated with various concentrations of PS14 (5, 10, 15, 20 and 25 µM) for 24 h, and examined under the inverted phase-contrast microscope. Further, the cells were stained with 2 µL of 10 µg/mL of Hoechst 33342 stain for 10 min in the dark [43]. The stained cells were washed in PBS and apoptotic cells were examined under a fluorescence microscope facilitated with a Cool SNAP-Pro color digital camera (Olympus, Tokyo, Japan). Results were recorded as photomicrographs.

#### 3.3.5. DCFH-DA

To determine the intracellular ROS production in PS14-exposed Hep-2 cells, the cell-permeable non-fluorescent DCFH-DA stain was utilized. In brief, 1 × 10^6^ cells per well were seeded in a six-well plate and allowed to reach the desired confluence. Then, the cells were treated with PS14 at different concentrations (5, 10, 15, 20 and 25 µM) for 24 h. As mentioned elsewhere, the standard protocol to measure ROS production was adopted [38].

#### 3.3.6. RT-PCR

The influence of PS14 to regulate the typical anti-cancer gene expressions, such as Bax, Bcl, Cas3 and Cas9, was evaluated, as described previously [44]. Total RNA was isolated using the standard trizol method. cDNA synthesis was carried out by following the manufacturer’s protocol (cDNA synthesis kit, Sigma). Light Cycler 96 (Roche Applied Science, Switzerland) and SYBR Premix ExTaq (Takara, Dalian, China) were used to perform reactions in a 10 μL volume that included 1 μL of cDNA, 5 μL of SYBR green master mix, 1 μL of primer mix and 3 μL of RNase-free water per reacting mixture. The qPCR thermal profile was set as 10 min at 95 °C, followed by 40 cycles of ten seconds at 95 °C, twenty seconds at 60 °C, and ten seconds at 72 °C. Similar thermal profile was used for the house-keeping molecule, GAPDH [44]. Primers used in this study are listed in Table 2. Gene expression data is represented in fold change, as per the 2^−ΔΔct^ method.

### 3.4. Assessment of In Vitro Anti-Inflammatory Activity

#### 3.4.1. Effect of PS14 on Proteinase Inhibitory Assay

Trypsin inhibition assay indicates the potential of PS14 to inhibit proteinases and mitigate tissue damage. This assay evaluated the extent of PS14-mediated protection against tissue damage during inflammation due to inhibition of proteinase production. The reaction mix contained (200 µL) 0.06 mg trypsin, (100 µL) 20 mM Tris HCl buffer (pH 7.4) and (100 µL) of PS14 (at various concentrations: 5, 10, 15, 20 and 25 µM) [45]. The mix was incubated at 37 °C for 5 min; after the 5 min incubation, (100 µL) 0.8 percent (*w*/*v*) bovine serum albumin was added. The above mixture was incubated for an additional 20 min. The reaction was eventually stopped by the addition of 5% trichloroacetic acid (TCA). The cloudy suspension was centrifuged for 5 min at 2500 rpm. Finally, using a UV-visible spectrophotometer, the absorbance of the supernatant was measured at 217 nm against a blank. A 100 µM diclofenac was used as a control.

#### 3.4.2. Effect of PS14 on Lipoxygenase Inhibition

Lipoxygenase enzyme is associated with the cellular pro-inflammatory responses via the generation of leukotrienes, the key pro-inflammatory molecules. Inhibition of lipoxygenase could mitigate the production of leukotrienes, which could have an anti-inflammatory effect. In the assay, linoleic acid was used as a substrate, and lipoxidase was used as an enzyme to test the anti-lipoxygenase activity [46]. The reaction mix consisting of 160 μL sodium phosphate buffer (100 mM; pH 8.0), 10 μL PS14 at various concentrations (5, 10, 15, 20, 25 µM and 100 µM) and 20 μL soybean lipoxygenase solution (167 U/mL) was incubated at 25 °C for 10 min. A 10 μL sodium linoleic acid solution was added to the substratum and initiated the reaction. The reaction was measured using a UV-visible spectrophotometer at 234 nm. A 100 µM diclofenac was used as a control.

#### 3.4.3. Effect of PS14 on Heat-Induced Hemolysis

The effect of PS15 on heat-induced hemolysis of human red blood cells (HRBC) was investigated. In brief, 5 mL of whole blood was collected from a healthy human volunteer. The collected blood was centrifuged at 3000 rpm for 10 min and washed three times in 10 mM sodium phosphate (154 mM NaCl) buffer (pH 7.4) to isolate the red blood cells. In the erythrocytes, PS14 at different concentrations were added, mixed gently and incubated in a water bath at 60 °C for 20 min. The centrifuge tubes were cooled under running tap water and centrifuged again at 3000 rpm for 5 min. Then, using the supernatant, the heat-induced hemolysis was measured at 560 nm using a UV-vis spectrophotometer [45].

#### 3.4.4. Effect of PS14 on Hypotonicity-Induced Hemolysis

Hypotonicity-induced hemolysis was performed using 50 mM NaCl hypotonic solution in 10 mM sodium phosphate buffer saline (pH 7.4) along with erythrocytes. A reaction mixture, whose final volume was 200 µL, was prepared with 100 µL of PS14 (at various concentrations) and 10% erythrocyte suspension in 100 µL. The above mixture was incubated for 30 min at 37 °C and then centrifuged for 20 min at 3000 rpm. A 100 µM diclofenac was used as a control. A UV-vis spectrophotometer set to 560 nm was used to measure the hemoglobin content of the supernatant [45].

### 3.5. Statistical Analysis

Data in the study are represented as the mean of three replicates ± standard deviation. One-way ANOVA and Tukey’s Multiple Range Test using Graph Pad Prism (Ver.5.0) were utilized to determine the statistical significance at 5%.

## 4. Conclusions

The present study concluded that the PS14 peptide derived from the CBD protein of *A. invadans* showed a potential anti-inflammatory and anticancer activity in both in vitro and in vivo. Moreover, the upregulation of ROS and apoptosis condition in the Hep-2 cells by PS14 peptide proves the anti-cancer activity. Therefore, we suggest that PS14 peptide can be considered as a novel treatment strategy for cancer. Further research needs to be performed in animal models to understand their effect on molecular signaling pathways.

## Figures and Tables

**Figure 1 molecules-27-07333-f001:**
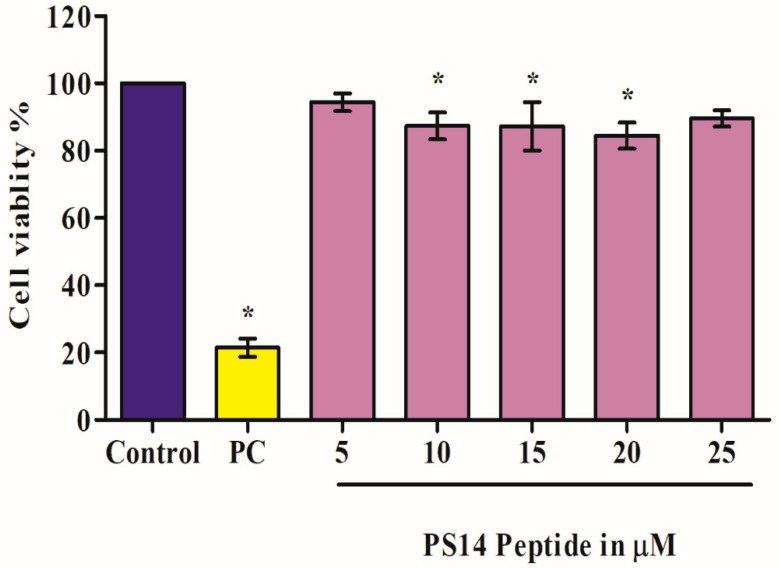
In vitro toxicity of PS14 tested on rat skeletal myoblast cells (L6 cell line) assessed by MTT assay. The bar shows the toxicity values of control (untreated), PC (positive control) Triton X-100 (0.01%) and PS14-treated experimental sample at five different concentrations. The asterix (*) denotes *p* < 0.05 level of significance compared to the control. Data were presented as mean ± standard deviation (SD) of three independent experiments.

**Figure 2 molecules-27-07333-f002:**
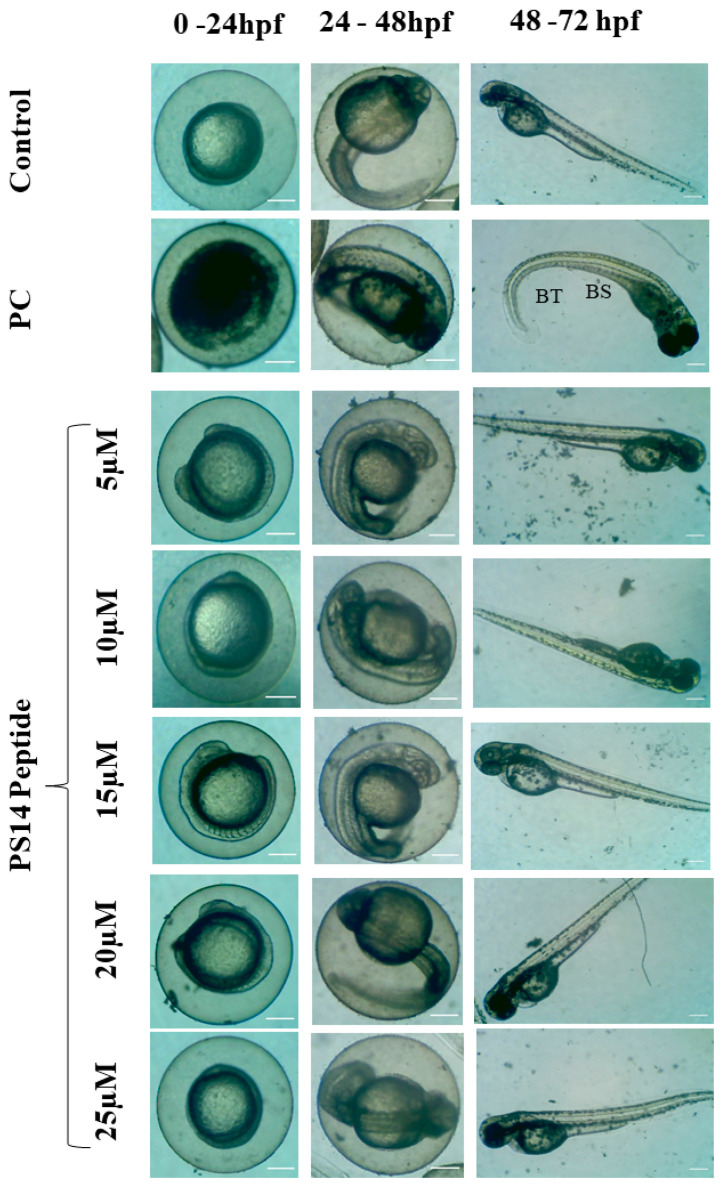
Developmental toxicity on zebrafish larvae observed between 0 hpf and 72 hpf. The larvae were treated with five different concentrations of PS14 along with an untreated control and positive control (PC) 1 mM of H_2_O_2_. PC showed abnormal morphologies such as bent spine (BS) and bent tail (BT).

**Figure 3 molecules-27-07333-f003:**
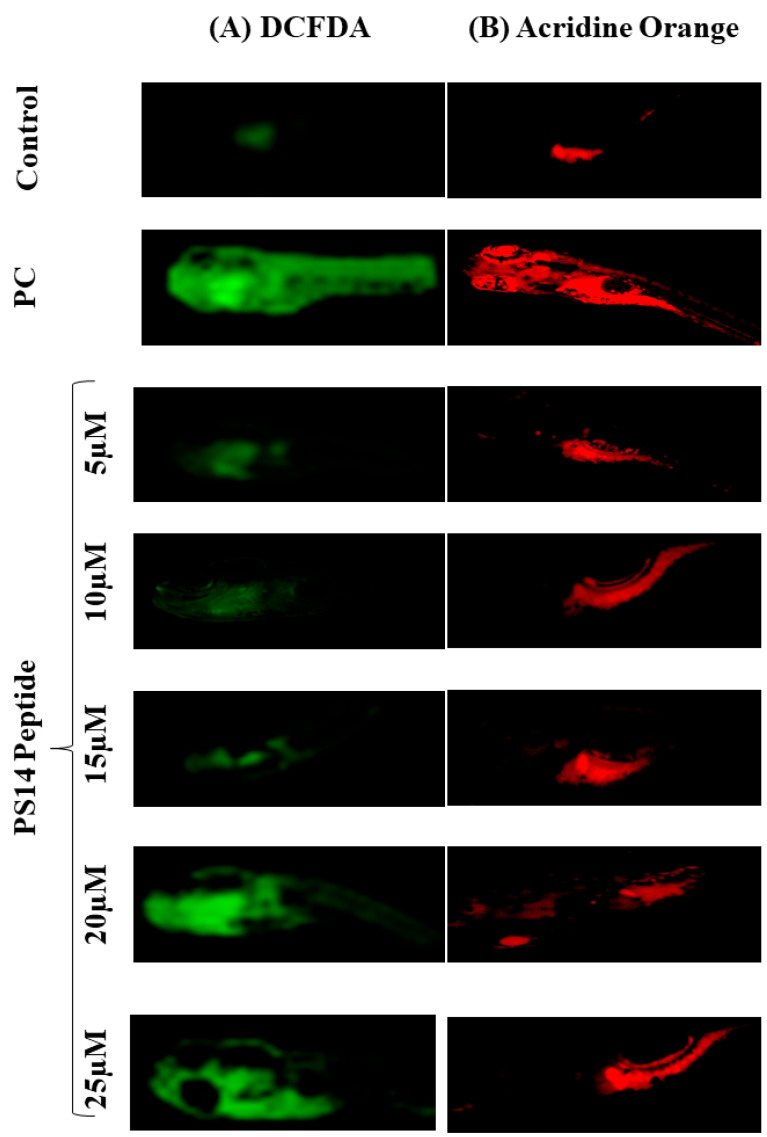
Fluorescent imaging of 96 hpf zebrafish larvae treated with PS14 along with untreated control and positive control (PC) treated with 1mM of H_2_O_2_. (**A**) Intercellular ROS measurement by DCFDA; and (**B**) Apoptosis detected by acridine orange.

**Figure 4 molecules-27-07333-f004:**
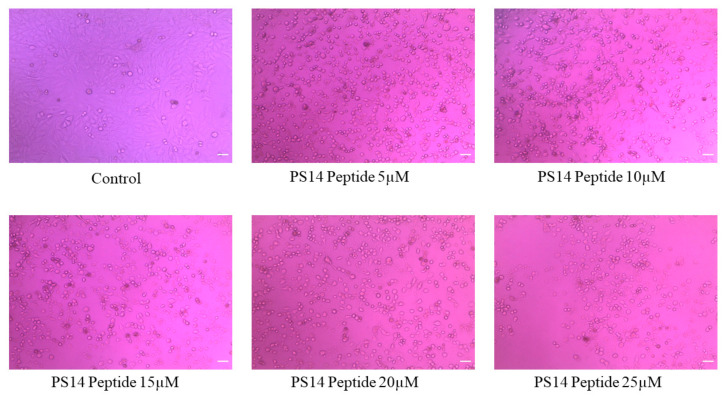
The morphological examination of PS14-treated Hep-2 cells observed under inverted Phase-contrast microscopic at 20× magnifications; untreated cells used as control.

**Figure 5 molecules-27-07333-f005:**
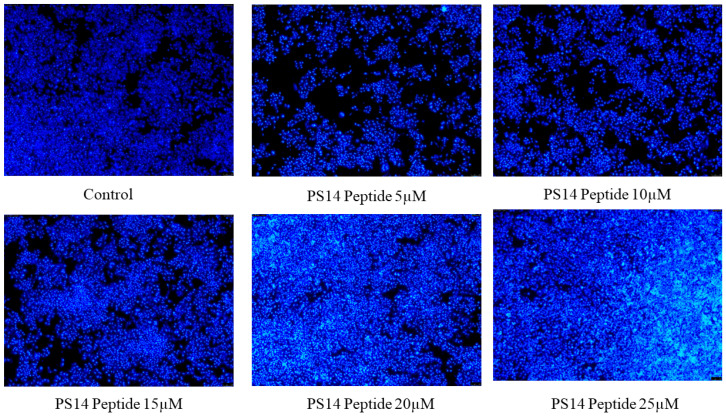
The Hoechst 33342 staining detected apoptotic induction in PS14-treated Hep-2 cells. The apoptotic cells were observed as high intensity of bright blue. No apoptotic induction was observed in the untreated control.

**Figure 6 molecules-27-07333-f006:**
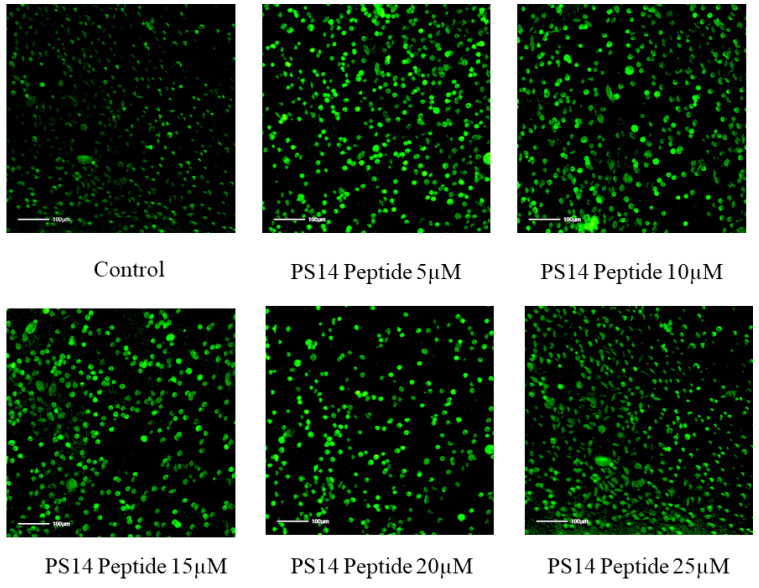
Measurement of the ability of PS14 intracellular ROS generation against Hep-2 cells. The ROS level was compared with the control (untreated) group.

**Figure 7 molecules-27-07333-f007:**
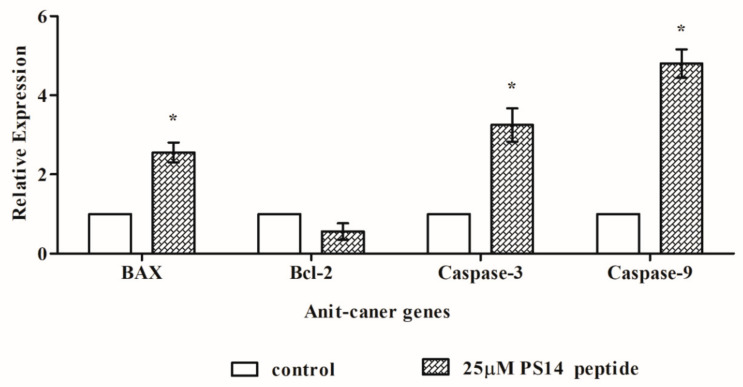
qPCR analysis of apoptotic gene expression due to PS14. The data were normalized with the internal control molecule, GAPDH. Results were compared with the untreated control and presented the data as fold change. The data were presented as mean ± SD of three replicates. The asterix (*) represents the significant difference at *p* < 0.05 compared to the control.

**Figure 8 molecules-27-07333-f008:**
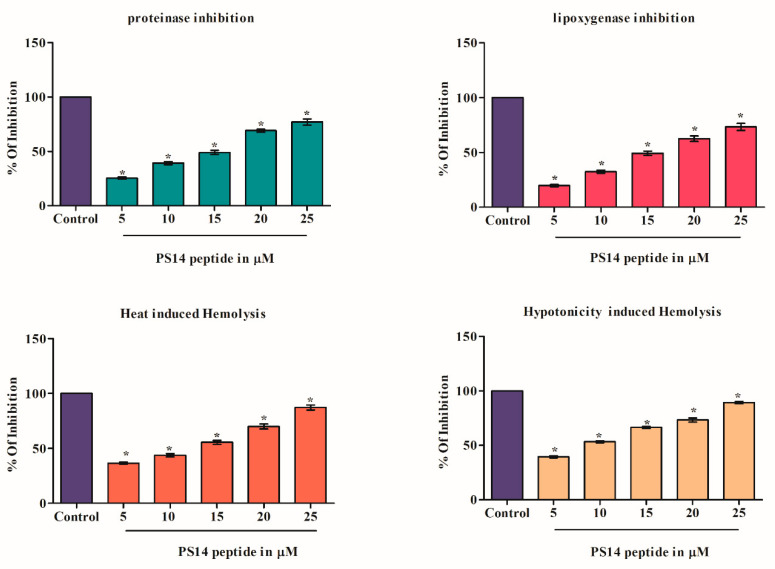
Anti-inflammatory activity of PS14. The results were compared with the untreated control. The data were presented as mean ± SD of three replicates. The asterix (*) represents the significant difference at *p* < 0.05 compared to the control.

**Table 1 molecules-27-07333-t001:** In vitro anti-cancer activity of PS14 against Hep-2 cells. Anti-cancer activity was determined based on the results of MTT, Trypan blue exclusion and LDH release assays. Control represents untreated group.

Concentration	Percentage of Inhibition		Cell Count Percentage at 24 h		Percentage of LDH Release at 24 h
	24 h	48 h	Viable	Dead	
**Control**	5.2 ± 0.25	7 ± 0.35	96.25 ± 1.32	4.28 ± 1.42	10.25 ± 0.25
**PS14 5 µM**	21.32 ± 3.2 *	19.22 ± 1.5 *	79.35 ± 0.94 *	23.86 ± 0.98 *	9.74 ± 1.54
**PS14 10 µM**	27.25 ± 1.89 *	25.35 ± 0.92 *	73.24 ± 4.25 *	28.78 ± 2.3 *	19.31 ± 1.09 *
**PS14 15 µM**	34.37 ± 2.45 *	39.93 ± 3.24 *	61.52 ± 3.8 *	36.48 ± 0.80 *	34.31 ± 1.6 *
**PS14 20 µM**	48.70 ± 2.1 *	58.60 ± 1.7 *	54.75 ± 4 *	48.25 ± 1.32 *	45.56 ± 2 *
**PS14 25 µM**	59.75 ± 1.7 *	57.41 ± 1.8 *	41.25 ± 5 *	66.32 ± 2.45 *	69.05 ± 1.6 *

* denotes *p* < 0.05 level of significance compared to the control. Data were presented as mean ± SD of three independent experiments.

**Table 2 molecules-27-07333-t002:** List of primers used for the anti-cancer gene expression study.

Gene	Primer Sequence	Reference
** *Bcl-2* **	Forward: 5′-GTGGATGACTGAGTACCT-3′ Reverse: 5′-CCAGGAGAAATCAAACAGAG-3′	[44]
** *BAX* **	Forward: 5′-TCAGGATGCGTCCACCAAGAAG-3′ Reverse: 5′-TGTGTCCACGGCGGCAATCATC-3′	
** *Caspase-3* **	Forward: 5′-ACATGGAAGCGAATCAATGGACTC-3′ Reverse: 5′-AAGGACTCAAATTCTGTTGCCACC-3′	
** *Caspase-9* **	Forward: 5′-GCTCTTCCTTTGTTCATC-3′ Reverse: 5′-CTCTTCCTCCACTGTTCA-3′	
**GAPDH (internal control)**	Forward: 5′-GTCTCCTCTGACTTCAACAGCG-3′ Reverse: 5′-ACCACCCTGTTGCTGTAGCCAA-3′	

## Data Availability

The datasets used and analyzed in the current study are available from the corresponding author upon reasonable request.

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
