# Peer review of "Anti-Cancer and Anti-Inflammatory Activities of a Short Molecule, PS14 Derived from the Virulent Cellulose Binding Domain of Aphanomyces invadans, on Human Laryngeal Epithelial Cells and an In Vivo Zebrafish Embryo Model"

_molecules, 2022, doi:10.3390/molecules27217333_

Round 1

Reviewer 1 Report

The manuscript is well presented. However, there are several things need to be explained:

1. Why Hep-2 cell line was used for the anticancer activity?

2. No positive control was used in the experiments to compare the activity of the peptide with it.

3. The mode of cell death could have been confirmed by performing annexin v/PI or caspase 3/7 induction experiments.

4. Why cell cycle analysis experiment was not used? It would provide some information about the possible mechanism of action. 

Reviewer 2 Report

This manuscript by Velayutham et al. describes “Anti-cancer and anti-inflammatory activities of a short molecule, PS14 derived from the virulent cellulose binding domain of Aphanomyces invadans on human laryngeal epithelial cells, and in-vivo zebrafish embryo model”. Although this short peptide did not display very potent anticancer activity, but authors have systematically explored it’s in vitro and in vivo studies. Introduction, results and discussion are written well in this manuscript. I would recommend this to be published after below minor corrections.

1.    Conclusion part can be improved

2.   Please upload all E.Supp.figures in supporting information data.

Author Response

Plz find attachment

Round 2

Reviewer 1 Report

There are no comments.